# A Green and Effective Polyethylene Glycols-Based Microwave-Assisted Extraction of Carnosic and Rosmarinic Acids from *Rosmarinus officinalis* Leaves

**DOI:** 10.3390/foods12091761

**Published:** 2023-04-24

**Authors:** Chunyan Zhu, Yunchang Fan, Xiujun Bai

**Affiliations:** 1College of Chemistry and Chemical Engineering, Henan Polytechnic University, Jiaozuo 454003, China; 212012020034@home.hpu.edu.cn; 2Shijiazhuang ENN Gas Co., Ltd., Shijiazhuang 050081, China; baixiujun@enn.cn

**Keywords:** antioxidant activity, green solvents, microwave-assisted extraction (MAE), *Rosmarinus officinalis* leaves (ROLs), carnosic acid (CA), rosmarinic acid (RA)

## Abstract

*Rosmarinus officinalis* leaves (ROLs) are widely used as a popular culinary spice for flavoring food, in which carnosic acid (CA) and rosmarinic acid (RA) are the main active components. The extraction of CA and RA is limited by lowextraction efficiency and extraction rate. In this work, a microwave-assisted extraction (MAE) method using biodegradable, low-toxic and nonflammable solvents polyethylene glycols (PEGs) as extraction solvents was developed for theextraction of CA and RA from ROLs. Experimental results suggest that PEG-400 was a better choice compared with PEG-200, and the optimal extraction conditions were as follows: 45% of PEG-400, 4.3% of phosphoric acid, 20 s of microwave irradiation time at 280 W of microwave irradiation power, and a 10 mg mL^−1^ solid–liquid ratio, respectively. The tissue structures of ROLs could be effectively disrupted by PEG-based MAE, leading to high CA and RA extraction efficiencies. The PEG-400 extract exhibited stronger 1,1-diphenyl-2-picrylhydrazyl (DPPH) radical scavenging ability compared with butylated hydroxytoluene (BHT). Finally, compared with heating reflux extraction, ultrasound-assisted extraction, maceration, and MAE using ionic liquid and ethanol as extraction solvents, the developed PEG-400 based MAE exhibited the highest extraction ability and fastest extraction rate for CA and RA. These findings suggest that MAE using PEGs as extraction solvents is a promising method for the separation of bioactive compounds from natural plants.

## 1. Introduction

*Rosmarinus officinalis* (RO), an evergreen shrub native to the Mediterranean region, is cultivated worldwide, and its leaves are commonly used both as a popular culinary spice for flavoring food and a traditional medicinal herb due to their powerful anti-inflammatory, anticancer, antioxidant, and antibacterial properties [1,2]. These biological activities can be ascribed to the facts that RO leaves (ROLs) are rich in antioxidant compounds, such as carnosic acid (CA) and rosmarinic acid (RA) [3,4]. The isolation of CA and RA from ROLs has become a hot research topic, and a variety of extraction techniques, including supercritical fluid extraction [3], maceration [5,6,7], heat reflux extraction (HRE) [5,8], ultrasound-assisted extraction (UAE) [4,9,10], and microwave-assisted extraction (MAE) [4,5,8], have been used for this purpose. Lefebvre and coworkers developed an online supercritical fluid extraction (SFE)–supercritical fluid chromatography system to selectively separate CA and RA from ROLs using carbon dioxide as extraction solvent and a mixture of ethanol and water as polar modifier, which obtained a 49% CA and 78% RA [3]. Although SFE is a green technique, the limitations of this method include the use of expensive instruments as well as the yield of lower purity crude products. Compared with SFE, maceration, HRE, UAE, and MAE are lower cost with easier to operate extraction methods. For example, Mazaud et al. used a 30% (wt%) *n*-butoxyethanol (BOE) aqueous solution (pH 2, adjusted by H_3_PO_4_) as extraction solvent to macerate ROLs at room temperature for 8 h to yield 1.02 g L^−1^ of CA. However, the extraction ability of BOE for RA is not mentioned in their work [6]. Oliveira et al. suggested a maceration method using 70% ethanol as extraction solvent. After macerating the ROLs at room temperature for 55 min, followed by drying the resultant extract at 110 °C to remove solvents, it obtained 3.92% (wt%, the percentage of the dry weight of extract) RA and 9.34% CA in the final extract [7]. Liu and coworkers developed an HRE method to extract CA and RA using 85% ethanol as extraction solvent with a 2 h refluxing time, which yielded an extraction efficiency of 28.3 mg g^−1^ and 3.86 mg g^−1^, for CA and RA, respectively [8]. Jacotet-Navarro et al. used the UAE technique to isolate CA and RA from ROLs with 90% ethanol as extraction solvent, and an extract containing 7.73% (wt%, the percentage of the dry weight of extract) of CA could be produced [9]. Very recently, Wang and coworkers constructed a thermoswitchable solvent system based on deep-eutectic solvents (DESs)/ionic liquid (IL, 1-butyl-3-methylimidazolium hexafluorophosphate ([C_4_mim]PF_6_))/water mixture to efficiently extract CA and RA from ROLs under ultrasonication for 20 min at 60 °C, at which the mixture of DESs + [C_4_mim]PF_6_ + water was a single-phase system. After cooling to 25 °C, this mixture was changed into a two-phase system with the upper phase as water containing DESs and RA and the lower phase as [C_4_mim]PF_6_ containing CA. For this method, the extraction efficiencies (weight of target compound/dry weight of ROL powder) of CA and RA were 17.74 mg g^−1^ and 7.06 mg g^−1^, respectively [10].

Generally, maceration, HRE, and UAE are time-consuming techniques, whereas MAE has the advantages of the fast extraction and low consumption of solvents. Sik et al. used MAE to extract RA from ROLs, and an extraction yield as high as 9.4 mg g^−1^ (weight of RA/dry weight of ROL powder) was obtained within 5 min [5]. Liu and coworkers suggested an MAE for the extraction of CA and RA from ROLs using the aqueous solution of the ionic liquid (IL) 1-octyl-3-methylimidazolium bromide ([C_8_mim]Br) (1.0 mol L^−1^) as extraction solvent, and this IL-based MAE exhibited higher extraction yield for CA and RA compared with conventional extraction methods such as HRE and hydrodistillation [8].

Despite progresses made in the extraction of RA and CA, there are still questions that need to be further investigated: (I) Although high extraction yields of RA and CA are obtained using the ILs [C_4_mim]PF_6_ and [C_8_mim]Br as extraction solvents, the two ILs are poorly biodegradable [11,12]. (II) Ethanol is a green solvent and is widely used in MAE [5], HRE [8], UAE [9], maceration [5,7], and SFE [3]; however, it is very volatile and highly flammable, which poses a potential risk for operators, especially when the extraction processes operate at high temperature and elevated pressure [13].

Based on the above discussion, it is urgent to find green, biodegradable, and nonflammable solvents to replace ILs and ethanol for the extraction of CA and RA. Therefore, this work aims to develop a green extraction technique based on the rapid extraction method MAE using the biodegradable, low-toxic, and nonflammable solvents polyethylene glycols (PEGs) [14,15,16,17,18] as extraction solvents.

## 2. Materials and Methods

### 2.1. Reagents and Materials

2,2-Diphenyl-1-picrylhydrazyl (DPPH, ≥97%), butylated hydroxytoluene (BHT, >99%), CA (≥97%), RA (≥97%), and BOE (99%) were purchased from Aladdin Biochemical Technology Co., Ltd. (Shanghai, China). Food-grade polyethylene glycol 200 (number-average molecular weight) (PEG-200) and PEG-400 were supplied by Yantian Biotechnology Co., Ltd. (Shanghai, China). The IL [C_8_mim]Br (99%) was obtained from Lanzhou Institute of Chemical Physics of the Chinese Academy of Sciences (Lanzhou, China). All the other reagents are analytical-grade and used as received. Dried ROLs (Severn Sea rosemary, harvested in October 2022 from Shandong Province, China) were supplied by Haoruijia Biotechnology Co., Ltd. (Chengdu, China) and crushed into fine powers (100 mesh) by a disintegrator (model: 08A1, Xulang Co., Guangzhou, China) before use. The moisture content of the ROL powder was 5.9% (wt%, dry weight basis), which was determined via the oven-dry method (105 °C, 24 h) [19,20].

### 2.2. Determination of CA and RA

The analysis of CA and RA was conducted by high performance liquid chromatography (HPLC, model: Agilent 1200, Agilent Technologies, Santa Clara, CA, USA) and the chromatographic conditions were as follows: separation column, Amethyst C18-H column; mobile phase, the mixture of acetonitrile and acetic acid (HAc) aqueous solution (HAc content: 0.1% (*v*/*v*)) (65% (*v*/*v*) of acetonitrile) with a flow-rate of 1.0 mL min^−1^ for the analysis of CA and the mixture of 20% (*v*/*v*) acetonitrile and 80% HAc aqueous solution (HAc content: 0.1% (*v*/*v*)) with a flow-rate of 0.85 mL min^−1^ for the determination of RA; detection wavelengths, 330 nm for analysis of RA and 250 nm for determination of CA; column temperature, 30 °C; and injection volume, 5.0 μL.

### 2.3. Extraction of CA and RA from ROLs

#### 2.3.1. The Developed MAE Method

Generally, PEG-200 and PEG-400 are widely used as solvents [18], and their extraction ability for CA and RA was thus studied in order to select the best extraction solvent. For a typical MAE procedure, 50 mg of ROL powder and 5.0 mL of PEG aqueous solution (PEG-200 or PEG-400) were put into a Teflon tube, and the resultant mixture was irradiated by a household microwave oven (model G70F20CPIII-TK(W0), Galanz Microwave Oven and Electrical Appliances Manufacturing Co., Ltd., Foshan, China). The MAE conditions are as follows: PEG concentration (*C*_PEG_, PEG-200 or PEG-400): 30% (*v*/*v*) to 90%; microwave irradiation power: 70 W to 420 W; microwave irradiation time: 5 s to 90 s; solid-liquid ratio: 5 mg mL^−1^ to 30 mg mL^−1^; and phosphoric acid (H_3_PO_4_) concentration in PEG solution: 0% (*w*/*v*, g mL^−1^) to 8.7%. After extraction, the extract was diluted to 10 mL with extraction solvent and subsequently centrifuged (TG16-WS high-speed centrifuge, Yuhua Instrument Co. Ltd., Gongyi, China) for 2 min. The resultant supernatant was diluted 10-fold with 50% (*v*/*v*) ethanol aqueous solution and filtered through a 0.45μm membrane before injecting into the HPLC system for the analysis of CA and RA. The extraction efficiency (*EE*%) was calculated using the following equation:(1)$$EE\%=\frac{\mathrm{weight}\;\mathrm{of}\;\mathrm{CA}\;\mathrm{or}\;\mathrm{RA}\;\left(g\right)}{\mathrm{dry}\;\mathrm{weight}\;\mathrm{of}\;\mathrm{ROL}\;\mathrm{powder}\;\left(g\right)}\times 100$$

#### 2.3.2. Reported Extraction Methods

##### The MAE Methods

The MAE method using 1.0 mol L^−1^ of [C_8_mim]Br aqueous solution as extraction solvent was carried out according to the reported procedure: the mixture of 1.0 g of ROL powder and 12.0 mL of [C_8_mim]Br aqueous solution at the concentration of 1.0 mol L^−1^ was irradiated by microwave for 15.0 min at a 700 W of microwave irradiation power (MDS-6G microwave extractor, Sineo Microwave Chemistry Technology Co., Ltd. (Shanghai, China)) [8].

The MAE with the mixture of ethanol, water, and HCl (70:29:1, *v*/*v*) as extraction solvent was conducted on the MDS-6G microwave extractor under 80 °C for 5.0 min with a solid–liquid ratio of 1:40 (i.e., 0.25 g ROL powder versus 10 mL extraction solvent) [5].

##### The UAE Method

The UAE procedure was conducted at 40 °C for 30 min using 20 mL of 90% (*v*/*v*) ethanol (extraction solvent) to extract 1.0 g of ROL powder in an ultrasound bath (1740QT ultrasonic cleaner, Kexi Technology Co., Ltd. (Beijing, China)) [9].

##### The HRE Method

The mixture of ROL powder (1.0 g) and 10 mL of 85% (*v*/*v*) ethanol aqueous solution was allowed to reflux for 2 h [8].

##### Maceration Methods

Maceration using 30% (wt%) of BOE aqueous solution (pH 2, adjusted by H_3_PO_4_) as extraction solvent was conducted by mixing 1.0 g of ROL powder with 10.0 mL of extraction solvent at 25 °C for 8 h under stirring [6].

Maceration with 70% (*v*/*v*) ethanol aqueous solution as extraction solvent was carried out by mixing 1.0 g of ROL powder with 5.0 mL of extraction solvent at 25 °C for 55 min under stirring [7].

### 2.4. Measurements of the DPPH Radical Scavenging Activity of the PEG-400 Extract

The DPPH radical scavenging activity of the PEG-400 extract was determined following the reported methods [21,22]: different volumes of the PEG-400 extract were mixed with 4.0 mL of DPPH (1.0 × 10^−4^ mol L^−1^ in methanol) and a given volume of methanol (total volume is set at 6.0 mL). The resultant mixture was stirred at room temperature for 30 min in the dark. The absorbance of mixtures was measured by a TU-1810 spectrophotometer (Persee Co., Beijing, China) at 517 nm, and BHT was used as the reference. The DPPH radical scavenging activity was calculated using the following equation [21,22]:(2)$$\mathrm{Scavenging}\;\mathrm{activity}\;(\%)=\frac{A_{\mathrm{control}}-A_{\mathrm{sample}}}{A_{\mathrm{control}}}\times 100$$

The viscosities of PEG-200 and PEG-400 solutions were determined by an NDJ-8S rotary viscometer (Changji Geological Instrument Co., Ltd., Shanghai, China). The morphologies of ROLs before and after PEG-based MAE were assessed by scanning electron microscopy (SEM, Quanta 250 FEG, Thermo Fisher Scientific, Hillsboro, OR, USA).

### 2.5. Experimental Design Using Response Surface Methodology (RSM)

Based on the results of single-factor experiments, the effects of variables, including PEG-400 concentration, H_3_PO_4_ concentration, microwave irradiation time, and microwave irradiation power, on the yields of CA and RA were investigated using a four-factor, three-level Box–Behnken design with RSM. Each factor (independent variable) was defined at three levels, namely low level (coded by −1), center level (coded by 0), and high level (coded by 1), as listed in Table 1. Second-order polynomial equations describing the relationship between independent variables and responses (yields of CA and RA) were used to fit the experimental data via a multiple regression procedure.

### 2.6. Statistical Analyses

Analyses of variance (ANOVA) and *t*-statistic of the experimental data were performed using Minitab 17 software (Minitab Inc., State College, PA, USA). A factor with a confidence level greater than 95% (*p* < 0.05) was considered to have a significant effect on the yields of CA and RA.

All experiments were conducted in duplicate, and the data are expressed as means ± standard deviations (SDs).

## 3. Results and Discussion

### 3.1. Selection of the Extraction Solvents

In the present work, PEG-200 and PEG-400 were used as extraction solvents, and their extraction ability for CA and RA is shown in Figure 1. It can be seen that when PEG-200 is used as extraction solvent, the extraction efficiency of CA increases with the increaseinPEG-200 concentration from 0% to 60%, remains stable from 60% to 80%, and decreases when further increasing to 90%, indicating that 60% of PEG-200 provides the highest CA extraction efficiency. A similar trend can be observed when using PEG-400 as extraction solvent: the extraction efficiency of CA increases with increasing the concentration of PEG-400 from 0% to 45%, stabilizes in the range of 45% to 60%, and decreases with further increasing the PEG-400 concentration. Therefore, a 45% PEG-400 yields the highest CA extraction efficiency, which is higher than the extraction ability of 45% PEG-200 and equivalent to the extraction ability of a 60% PEG-200, which may be attributed to the hydrophobic nature of CA [8,10] and the fact that PEG-400 is more hydrophobic than PEG-200 [23,24], i.e., the hydrophobic interaction between PEG-400 and CA is stronger than that between PEG-200 and CA, leading to PEG-400 having higher extraction ability for CA.

As far as the extraction of RA is concerned (Figure 1), its extraction efficiency increases with the increasing concentration of PEG-200 from 0% to 30% and decreases with further raising PEG-200 concentration. When PEG-400 is used as extraction solvent, the extraction efficiency of RA increases from 0% to 45% and decreases above this level. This suggests that 30% PEG-200 and 45% PEG-400 give the highest RA extraction efficiency, with the former having higher extraction ability than the latter. It is known that RA is a hydrophilic acid [8,10] and PEG-200 is more hydrophilic than PEG-400, meaning that the affinity between RA and PEG-200 is stronger than that between RA and PEG-400. Therefore, PEG-200 has higher extraction ability for RA compared with PEG-400.

It is known that PEG has good solubility for organic compounds [25], with lower PEG concentrations enhancing higher extraction efficiencies of CA and RA, while high PEG concentrations (e.g., 90%) are unfavorable, which may be ascribed to the highly viscous properties of PEG. As illustrated in Figure 2, the viscosity of the PEG solution increases with the increase in PEG concentration. High viscosity means low mass transfer rate, which consequently leads to low extraction efficiency [26,27].

The above findings indicated that 60% and 45% are the optimal PEG-200 and PEG-400 concentrations, respectively, to yield the highest extraction efficiencies for CA and RA. Since 45% PEG-400 can give similar CA and RA extraction efficiencies and consumes less PEG compared with 60% PEG-200, it was therefore selected as extraction solvent in the following experiments.

### 3.2. Effects of H_3_PO_4_Concentration, Microwave Irradiation Time, Microwave Irradiation Power, and Solid–Liquid Ratio on the Extraction Efficiencies of CA and RA

Generally, for the extraction of bioactive compounds from plant tissues, the addition of mineral acids can enhance the extraction efficiency, because they can effectively disrupt the cell walls of plant tissues [28,29]. Furthermore, it has been reported that H_3_PO_4_ can stabilize CA [6], and the effect of H_3_PO_4_ dosage is therefore studied. The results illustrated in Figure 3a suggest that the extraction efficiency of CA increases with increasing H_3_PO_4_ concentration from 0% to 4.3% and remains roughly stable when above this level. The extraction efficiency of RA keeps constant with the H_3_PO_4_ concentration, varying from 0% to 8.7%. Based on this observation, 4.3% was selected as the optimal H_3_PO_4_ dosage and used in the following experiments.

As shown in Figure 3b, microwave irradiation time is also an important parameter affecting the extraction efficiency. The extraction efficiency of CA enhances with increasing the microwave irradiation time from 5 s to 20 s and then stabilizes when further extending the microwave irradiation time. The extraction efficiency of RA remains constant with the microwave irradiation time in the range of 5 s to 90 s. Therefore, 20 s is regarded as the optimal microwave irradiation time. It should be noted that the extraction time of CA and RA by IL-based MAE [8], ethanol-based MAE [5], HRE [8], UAE [9], and maceration [6,7] is usually in the range of 5 min to 8 h. Rapid extraction is one of the advantages of the suggested PEG-400-based MAE compared with the reported methods.

The effect of microwave irradiation power shown in Figure 4a demonstrates that the extraction efficiencies of RA and CA increase by raising the microwave irradiation power from 70 W to 280 W and decrease with increasing the microwave irradiation power to 420 W. Based on this observation, 280 W is regarded as the optimal microwave irradiation. It has been reported that RA and CA are thermally unstable: high temperature leads to their degradation [30,31]. Therefore, the decrease in the extraction efficiency at 420 W of microwave irradiation power may be ascribed to the fact that this microwave irradiation power gives the highest extraction temperature, as illustrated in Figure 4a. In addition, Figure 4b demonstrates that a solid–liquid ratio of 10 mg mL^−1^ possesses the highest extraction efficiency for both CA and RA, and it was thus selected in the subsequent experiments.

### 3.3. Analysis of RSM

According to the results of the single-factor experiments, the effects of variables affecting the extraction efficiencies of CA and RA were further evaluated by RSM by carrying out 25 runs of experiments using the Box–Behnken design (Table 2). To explore the relationship between independent variables and responses (yields of CA and RA), second-order polynomial equations were used to fit the experimental data, and the model equations are as follows:Y_1_ (yield of RA, %) = 1.34 − 0.014A − 0.016B + 0.016C + 0.046D + 0.0025AB + 0.018AC + 0.0058AD − 0.038BC − 0.036BD + 0.016CD − 0.026A^2^ + 0.0052B^2^ − 0.038C^2^ − 0.17D^2^(3)
Y_2_ (yield of CA, %) = 2.64 + 0.22A + 0.064B + 0.26C − 0.077D − 0.0078AB − 0.030AC − 0.16AD − 0.12BC − 0.034BD − 0.028CD − 0.42A^2^ − 0.17B^2^ − 0.45C^2^ − 0.21D^2^(4)

An analysis of variance (ANOVA) was employed to evaluate the statistical significance of the above models, and the results are listed in Table 3. The regression coefficients and the corresponding *p*-values and *t*-values for the regression models are presented in Appendix A; the contour plots showing the interactive effects of any two variables on the extraction of CA and RA are shown in Appendix A. As can be seen from Table 3 and Appendix A, the two fitted models are all significant due to their low *p* values (<0.05). For the extraction of CA, PEG-400 concentration (A, *p* < 0.05) and microwave irradiation time (C, *p* < 0.05) are the variables significantly affecting the extraction efficiency, with a *p* < 0.05 in the interaction term AD indicating a significant interaction between PEG-400 concentration and microwave irradiation power; the interactions between other variables are not significant (*p* > 0.05) (Table 3 and Appendix A). As far as the extraction of RA is concerned, the microwave irradiation power (D) has a significant effect on the extraction efficiency (*p* < 0.05), and the interactive effects of any two variables on the extraction efficiency of RA are not significant, as listed in Table 3 and Appendix A.

### 3.4. Microstructural Characteristics of ROLs

It is reported that MAE can effectively disrupt the ROL structures, leading to an effective extraction of CA and RA [8,30]. To explore the changes in the ROL structures, the ROL samples prior and post MAE were investigated by SEM, and the results illustrated in Figure 5 indicate that MAE can effectively disrupt the surface microstructures of the ROL samples, demonstrated by the concave and porous shapes on the surface of the samples post MAE, which could result in the rapid release of CA and RA from ROLs and consequently lead to high extraction efficiencies.

### 3.5. Antioxidant Activity of the Extract

Generally, the DPPH radical scavenging activity is usually used to evaluate the antioxidant activities of plant extracts [21,22]. As shown in Figure 6, the PEG-400 extract exhibits a 90% DPPH radical scavenging activity at a concentration of 6.3 mg L^−1^ (total concentration of CA and RA in the PEG-400 extract), which is more effective than BHT (41.7 mg L^−1^). Additionally, PEG-400 is biodegradable with low toxicity [14,15,16]; therefore, the PEG-400 extract has potential applications as an antioxidant in the food industry.

### 3.6. Comparison with the Reported Extraction Methods

Recently, IL- or ethanol-based MAE [5,8], HRE [8], UAE [9], and maceration [6,7] techniques were used to extract CA and RA from ROLs. The extraction capacity of the developed PEG-400-based MAE was thus compared with these extraction methods. In order to ensure comparability of the experimental results, all reported extraction methods were performed under their optimal conditions. The results illustrated in Figure 7 indicate that the developed PEG-400-based MAE yields the highest CA and RA extraction efficiencies with the shortest extraction time. In addition, PEG-400 is environmentally friendly, as it is biodegradable with low toxicity [15,16].

## 4. Conclusions

In this work, a green MAE technique using the biodegradable, low-toxic, and nonflammable solvent PEG as the extraction solvent for the isolation of CA and RA from *Rosmarinus officinalis* leaves was developed. The experimental results show that PEG-based MAE exhibited the highest extraction efficiencies and fastest extraction rate for CA and RA compared with MAE using ethanol and IL, UAE with 90% ethanol, HRE with 85% ethanol, and maceration with ethanol and BOE as extraction solvents. The presence of H_3_PO_4_ improved the CA extraction efficiency because it could effectively disrupt the plant cell walls. Extraction temperature was another key factor affecting the extraction ability of PEG due to the thermally unstable nature of CA and RA with high temperature, leading to the decomposition of CA and RA. The RSM analysis indicated that microwave irradiation power had a significant effect on the extraction of RA; PEG-400 concentration and microwave irradiation time significantly affected the extraction of CA. These results lead to the conclusion that PEG-based MAE is a simple, rapid, environmentally friendly, and efficient method for the extraction of CA and RA from *Rosmarinus officinalis* leaves.

## Figures and Tables

**Figure 1 foods-12-01761-f001:**
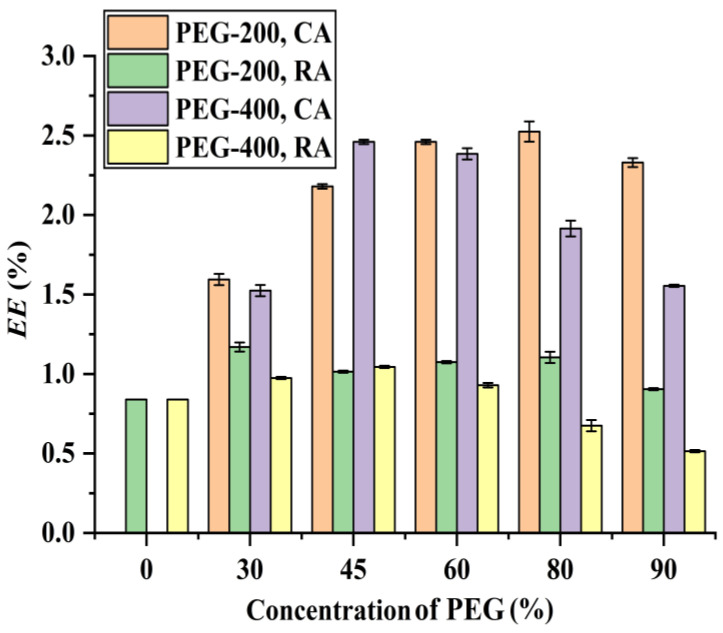
Effect of PEG concentration on the extraction of RA and CA:Solid–liquid ratio, 10 mg mL^−1^; microwave irradiation time, 40 s; microwave irradiation power, 210 W; and H_3_PO_4_ concentration, 4.3%.

**Figure 2 foods-12-01761-f002:**
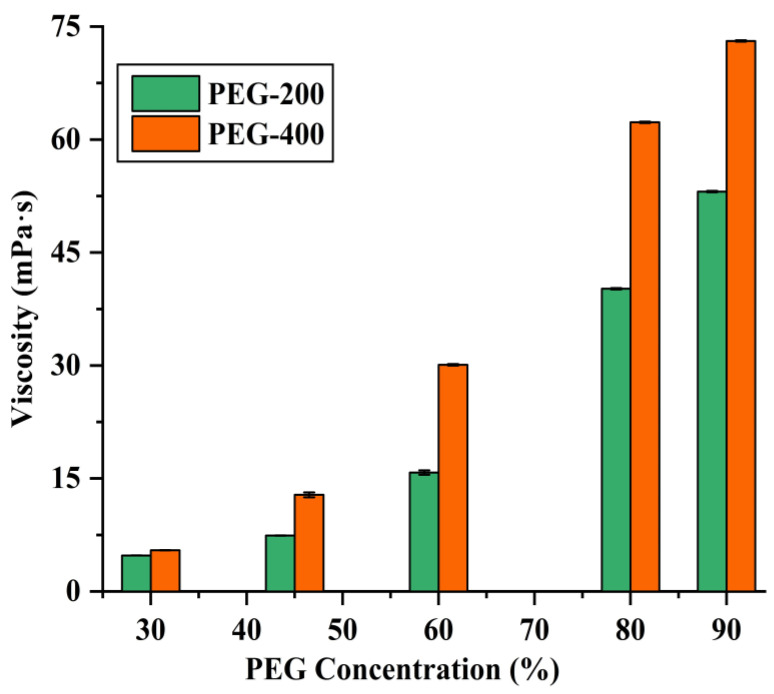
Viscosities of different PEG concentrations.

**Figure 3 foods-12-01761-f003:**
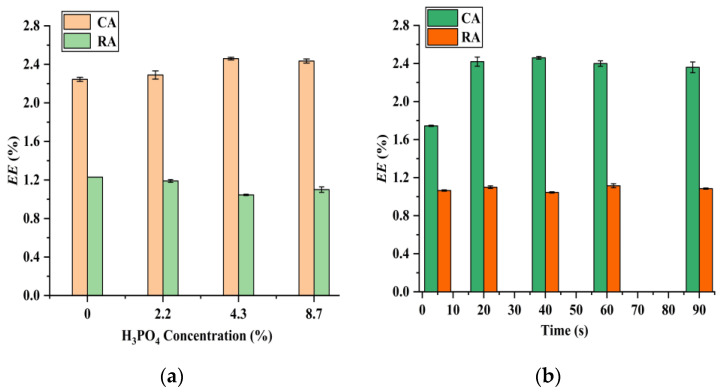
Effect of H_3_PO_4_ concentration (**a**) (solid–liquid ratio, 10 mg mL^−1^; microwave irradiation time, 40 s; microwave irradiation power, 210 W; and 45% PEG-400) and microwave irradiation time (**b**) (solid–liquid ratio, 10 mg mL^−1^; microwave irradiation power, 210 W; H_3_PO_4_ concentration, 4.3%; and 45% PEG-400) on the extraction of CA and RA.

**Figure 4 foods-12-01761-f004:**
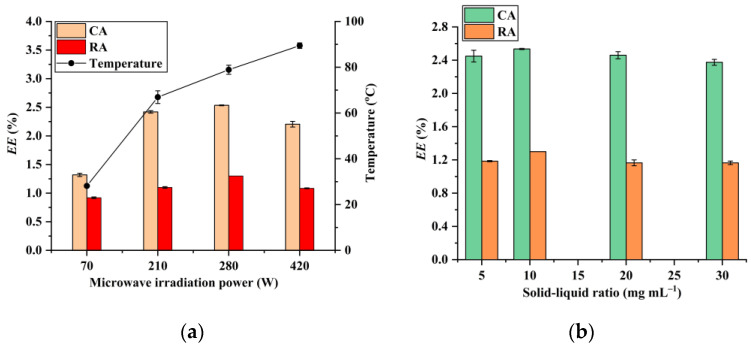
Effect of microwave irradiation power (**a**) (solid–liquid ratio, 10 mg mL^−1^; microwave irradiation time, 20 s; H_3_PO_4_ concentration, 4.3%; and 45% PEG-400) and solid–liquid ratio (**b**) (microwave irradiation time, 20 s; microwave irradiation power, 280 W; H_3_PO_4_ concentration, 4.3%; and 45% PEG-400) on the extraction of CA and RA.

**Figure 5 foods-12-01761-f005:**
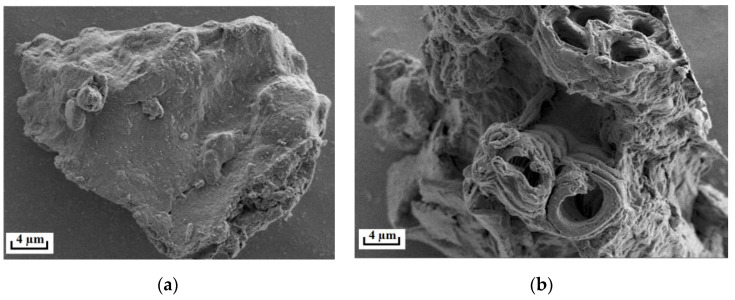
The SEM images of ROLs before (**a**) and after (**b**) PEG-400-based MAE.

**Figure 6 foods-12-01761-f006:**
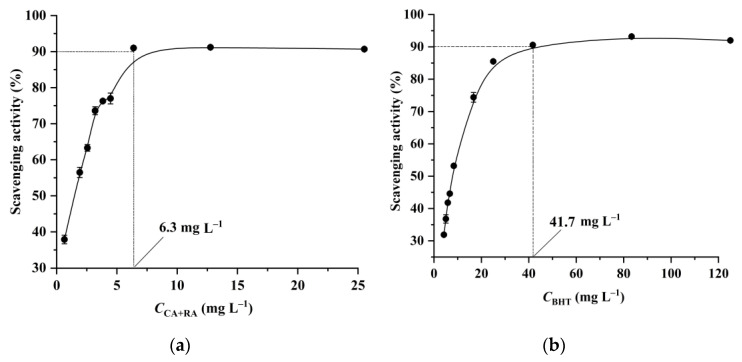
The DPPH radical scavenging activity of the PEG-400 extract (**a**) and BHT (**b**).

**Figure 7 foods-12-01761-f007:**
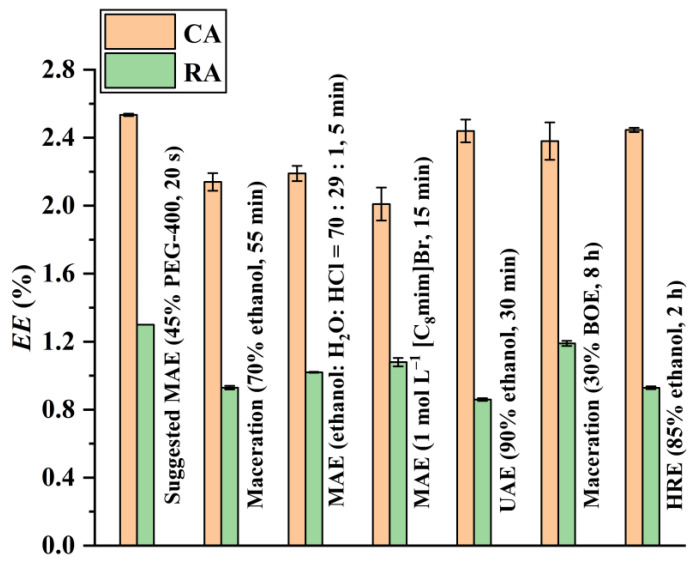
Comparison of the extraction ability ofCA and RA between the suggested PEG-400-based MAE and the reported methods.

**Table 1 foods-12-01761-t001:** Independent variables and their levels used in the Box–Behnken design.

Independent Variable	Coded Variable Level		
	Low Level (−1)	Center Level (0)	High Level (+1)
PEG-400 concentration (%), A	30	45	60
H_3_PO_4_concentration (%), B	2.2	4.3	8.7
Microwave irradiation time(s), C	5	20	40
Microwave irradiation power(W), D	210	280	420

**Table 2 foods-12-01761-t002:** The Box–Behnken experimental design for the extraction of CA and RA.

Run	A	B	C	D	Response (*EE*, %)	
					CA	RA
1	45	4.3	5	210	1.75	1.07
2	30	8.7	20	280	1.97	1.29
3	45	8.7	40	280	2.16	1.24
4	45	2.2	40	280	2.34	1.33
5	45	4.3	5	420	1.48	1.14
6	60	2.2	20	280	2.04	1.22
7	30	4.3	20	210	1.43	1.11
8	60	4.3	20	210	2.41	1.09
9	60	8.7	20	280	2.45	1.24
10	45	4.3	40	210	2.46	1.05
11	45	4.3	20	280	2.54	1.30
12	45	2.2	20	420	2.16	1.23
13	60	4.3	5	280	1.66	1.14
14	30	4.3	20	420	1.80	1.24
15	45	2.2	20	210	2.17	1.09
16	45	8.7	20	420	2.18	1.16
17	30	2.2	20	280	1.59	1.30
18	45	2.2	5	280	1.69	1.29
19	30	4.3	5	280	1.13	1.20
20	60	4.3	20	420	2.03	1.23
21	30	4.3	40	280	1.82	1.29
22	45	8.7	5	280	1.90	1.32
23	60	4.3	40	280	2.25	1.30
24	45	8.7	20	210	2.32	1.16
25	45	4.3	40	420	2.09	1.21

Note: A: PEG-400 concentration (%); B: H_3_PO_4_ concentration (%); C: microwave irradiation time (s); D: microwave irradiation power (W).

**Table 3 foods-12-01761-t003:** Analysis of variance (ANOVA) of the Box–Behnken design.

Source	SS	DF	MS	F-Value	*p*-Value	Remark
RA						
Model	0.15	14	0.011	5.32	0.006	Significant
A	1.737 × 10^−3^	1	1.737 × 10^−3^	0.85	0.378	
B	2.567 × 10^−3^	1	2.567 × 10^−3^	1.26	0.288	
C	2.443 × 10^−3^	1	2.443 × 10^−3^	1.20	0.300	
D	0.022	1	0.022	10.81	0.008	
AB	2.737 × 10^−5^	1	2.737 × 10^−5^	0.013	0.910	
AC	1.328 × 10^−3^	1	1.328 × 10^−3^	0.65	0.439	
AD	1.438 × 10^−4^	1	1.438 × 10^−4^	0.070	0.796	
BC	6.248 × 10^−3^	1	6.248 × 10^−3^	3.06	0.111	
BD	5.949 × 10^−3^	1	5.949 × 10^−3^	2.92	0.118	
CD	1.079 × 10^−3^	1	1.079 × 10^−3^	0.53	0.484	
A^2^	1.832 × 10^−3^	1	1.832 × 10^−3^	0.90	0.366	
B^2^	5.594 × 10^−5^	1	5.594 × 10^−5^	0.027	0.872	
C^2^	3.874 × 10^−3^	1	3.874 × 10^−3^	1.90	0.198	
D^2^	0.063	1	0.063	30.92	<0.0001	
Residual	0.020	10	2.040 × 10^−3^			
Cor Total	0.17	24				
CA						
Model	2.98	14	0.21	12.02	<0.0001	Significant
A	0.43	1	0.43	24.11	0.001	
B	0.042	1	0.042	2.37	0.155	
C	0.61	1	0.61	34.67	<0.0001	
D	0.061	1	0.061	3.46	0.092	
AB	2.662 × 10^−4^	1	2.662 × 10^−4^	0.015	0.905	
AC	3.625 × 10^−3^	1	3.625 × 10^−3^	0.20	0.660	
AD	0.11	1	0.11	6.33	0.031	
BC	0.064	1	0.064	3.63	0.086	
BD	5.257 × 10^−3^	1	5.257 × 10^−3^	0.30	0.598	
CD	3.486 × 10^−3^	1	3.486 × 10^−3^	0.20	0.666	
A^2^	0.50	1	0.50	28.08	<0.0001	
B^2^	0.060	1	0.060	3.40	0.095	
C^2^	0.55	1	0.55	30.92	<0.0001	
D^2^	0.093	1	0.093	5.26	0.045	
Residual	0.18	10	0.018			
Cor Total	3.15	24				

Note: SS: sum of squares; DF: degrees of freedom; MS: mean square; and *p*-values less than 0.05 are considered as statistically significant.

## Data Availability

Data are contained within the article.

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
