# Peer review of "A Green and Effective Polyethylene Glycols-Based Microwave-Assisted Extraction of Carnosic and Rosmarinic Acids from Rosmarinus officinalis Leaves"

_foods, 2023, doi:10.3390/foods12091761_

Round 1

Reviewer 1 Report

The study is concerned with rapidly extracting bioactive compounds, namely CA and RA, from Rosmarinus officinalis. This comprehensive study will impact the development of those bioactive compounds in the flavoring food industry. The author has delivered the results. However, several points must be improved . The suggestions are listed below:

Abstract: 

- the abstract did not cover all the study parameters, and the study about the antioxidant capacity using DPPH has not been mentioned yet

- line 11: the conjunction “and” between “flavoring food” and “carnosic acid” can be replaced with “which” to make the sentence clearer.

- Keywords: please consider using other keywords which different from the title to enhance the discoverability.

Material and methods:

- Line 111: more detailed information regarding the ROLs (type, harvesting period, etc) must be provided

- Section 2.3: Why did the concentration of BNNDESs in the water need to be analyzed? 

- Lien 208: please add space between the word “concentration” and “after”

Result and discussion:

- Section 3.1: the selection of the extraction solvents has not been explained in the material and methods

- Line 256: according to the previous explanation, the best PEG 200 to extract RA was 30% instead of 45%

- Line 294: which microwave power CA and RA gave the highest recovery?

- Line 299: the procedure of SEM analysis has not been mentioned yet in the material and methods

- Line 311: the word “better than” was doubled

- Line 312: Lack of discussion about the food safety of the obtained extract using PEG-400, please provide more information. Is the PEG-400 classified as food-grade?

Reviewer 2 Report

This is an interesting paper and would be useful to those looking for new and novel antioxidants. I have the following comments and suggestions for further improvement of the paper.

1. Line 49. What is the "high concentration of CA" obtained?

2. Lines 54,  the final extractions of RA & CA expressed as a percent of what? 

3. Line 56, the extraction efficiencies of RA & CA were expressed as mg per gram of what? dry wt of ROL? 

4. Line 58, is the CA extracted expressed as a percent of the extraction solution? 

5. Lines 66 and 70, are the extraction efficiencies expressed as mg per gram of dry wt of ROL?

6. Line 111, What was the moisture content of the dried ROL?

7. Line 116, Spelling mistake. Should be "denoted" not donated.

8. Lines 174- 179, why was the aqueous ethanolic concentration different (90% and 85%) in the two cases? and also the amount of ROL powder used (0.5g vs 1.0g)?

9. Figure 1, X-Axis, Concentration of PEG

10. Line 226-230, The meaning is lost in the construction of this long sentence. Break it into shorter sentences to bring out the meaning clearly.

11. Lines 235-236, Your statement that "Higher extraction ability of PEG 400 than PEG 200..." is not obvious from the graph.

12. Lines 264, "...mineral acids can efficiently...". delete 'efficiently'.

13. Figure 4b is invisible.

14. Line 299, "Some reported work has been reported that....." poor sentence construction.

15. Line 311, "better than" repeated twice.

16. There were several variables such as the concentration of PEG, Time, and Temperature. This experiment could have been designed better if you used Response Surface Methodology to study the effect of these variables on the extraction efficiency and their interactions. In the present study, only the effects of individual variables were studied, but not their interactions. Your study also lacks statistical validation of results.

Reviewer 3 Report

The aim of this work was to find alternative green solvents to replace ILs and ethanol for the extraction of carnosic acid and rosmarinic acid from Rosmarinus Officinalis leaves and to develop optimal extraction conditions for microwave-based green extraction technique and purification of carnosic acid and rosmarinic acid.

Preparation of ROL powder is not presented.

The results are very valuable. However,  the comparison with the reported extraction methods is not very satisfactory, because there are many different parameters and it is not possible to say with certainty what is the reason for the better extraction performances.

Reviewer 4 Report

The title of the manuscript is "A Green and Effective Polyethylene Glycols-Based Microwave-Assisted Extraction of Carnosic and Rosmarinic Acids from Rosmarinus Officinalis Leaves". However, when reading the paper, everything else that is done here completelly outweights the method mentioned in the title. The paper firstly mentions the effect of PEG concentration on the extraction yield, then moves to the effect of H3PO4, then includes microwave extraction, UAE, maceration, HRE, the effect of time, temperature, ethanol and more. Its contents is completely unconnected to the title and the presentation of data and methods is confusing, hard to follow and unclear. Furthermore, there was no experiment design used, it seems like all the experiments done were mere combinations of different conditions without a clear plan. I suggest the Authors choose a single part of their research (e.g. focusing only on PEG and MAE, as stated in the title) and show only this data. This requires a complete rewritting and restructuring of the paper, and a resubmission of a novel manuscript.

Round 2

Reviewer 1 Report

The manuscript has been improved and can be considered for a publication in Foods

Author Response

Thank you very much for your comments!

Reviewer 2 Report

The authors have addressed some of the questions that I raised, but I am not happy with their response to my question about the RSM for optimization of extraction of CA and RA. 

1. The Box Behnken experimental design with the range and level of variables was not given. It should be given under Materials and Methods.

2. What was the Model equation? Nothing was mentioned about the interactions of individual variables. 

3. Where are the response surface contour plots?

4. Under Materials and Methods, a separate section should be open giving how you did the statistical analyses, including ANOVA and t-statistic of the experimental data.
